# OpenReview forum: "InfoRM: Mitigating Reward Hacking in RLHF via Information-Theoretic Reward Modeling"
_NeurIPS.cc/2024/Conference — NeurIPS 2024 poster_

### Official Review · Reviewer_YUjH · 2024-06-12

**Soundness:** 2
**Presentation:** 2
**Contribution:** 2
**Rating:** 5
**Confidence:** 3

**Summary:**

This paper adds an information bottleneck objective for training a hidden representation and reward head in RLHF. They then empirically investigate advantages in policy optimization using such a reward model: there is less reward model overoptimization as judged by a gold RM, and overoptimization can be spotted by outlier detection in the latent space.

**Strengths:**

The main strength is the apparently strong empirical results, highlighting the potential practical applicability for frontier model training.

**Weaknesses:**

In my weaknesses, I fully focus on the theoretical analysis, hoping that other reviewers touch on scrutinizing the empirical investigations. Overall, the theoretical derivations contain numerous local mistakes. Sometimes, these mistakes "cancel out" again, leading to correct overall conclusions, but in one case, I remain skeptical or confused. There is a chance that the authors actually implemented everything correctly by good intuitions, but the theoretical investigations themselves are lacking in clarity and correctness. I now elaborate on this further.
*I am willing to raise my score if my theoretical concerns are fully addressed, though this potentially depends on how convinced the other reviewers are of the experiments.*

**1. Lower bound of $I(Y; S)$:** The authors give several *different* formulas for this lower bound:
- Equation (4): $I(Y; S) \geq E_{(x, y)}\Big[\int p_{\phi}(s, x) \log q_{\psi}(y \mid s) ds\Big]$.
- Equation (9): $I(Y; S) \geq \int p_{\phi}(x, s) \log q_{\psi}(y \mid s) ds dy$.
- Equation (27): $I(Y ; S) \geq \int p_{\phi}(s, x) \log q_{\psi}(y \mid s) ds$.

All three formulas are incorrect. The correct formula is:
- $I(S; Y) \geq E_{(x, y)} \Big[ \int p_{\phi}(s \mid x) \log q_{\psi}(y \mid s) ds \Big]$.

This can be verified by reading their reference [1], "Deep variational information bottleneck", Equation (12). Equation (4) is incorrect since it contains a factor $p(x)$ too much. Equation (9) is incorrect since it's not coherent: $x$ is not integrated over, but appears in the integrand. Same for (27): $x$ and $y$ appear in the integrand, but are not integrated over.

However, despite these many local mistakes, their end result, Equation (29), seems to contain the correct (!) lowerbound once everything is written down in concrete samples. However, this requires one more unstated assumption, namely that $p_{\phi}(s \mid x) = \delta_{f_{\phi}(x)}(s)$, i.e., that the distribution in the hidden state is deterministically the output of $f_{\phi}$. If this is assumed, then we have an explanation for why the integral over $S$ in Equation (29) disappears, and everything seems correct.

**2. Upper Bound of $I(X, S \mid Y)$:**
- The authors claim there is a Markov chain $X \to S \to Y$. This is incorrect: There is a Markov chain $S \to X \to Y$. It is obvious that the first Markov chain cannot hold in general since the probability of $S$ given $X$ is given by the encoder, so if the encoder removes ALL information, then no information about $Y$ can be left. I think the authors may have confused themselves by thinking about the Markov chain $X \to S \to \hat{Y}$, where $\hat{Y}$ is the RV of *outputs of the model* instead of the true data distribution.
- The authors derive locally incorrect conclusions from the Markov chain property in Equations (22) and (23).
- Equation (24) is also wrong in general: E.g., if the encoder deterministically maps each $x$ to the same point $s = \star$, then $I(X; S) = 0$, but the RHS would usually still be positive.

Despite these many mistakes, the authors somehow conclude with the correct (!) inequality $I(X; S \mid Y) \leq I(X; S)$ in Equation (25), and the rest of the derivation does not make use of the mistakes before anymore. Note, however, that the inequality in (25) is very elementary and well-known, and so does not need a one page long proof. One could either state it without proof or very easily find references for it (e.g., it's a sort-of dual of the data processing inequality and e.g. [follows directly from the proof given in wikipedia](https://en.wikipedia.org/wiki/Data_processing_inequality)). Though my true suggestion, actually, would be to just replace $I(X; S \mid Y)$ by $I(X; S)$ entirely since this is closer to the variational bound, increasing clarity on what is done in this paper.

In the final formula for the upper bound of $I(X;S \mid Y)$ in Equation (29), there appears the following KL divergence: $KL[f_{\phi}(x_w), r(S)]$. **I do not understand what this is:** As established before, to make sense of the lower bound of $I(S; Y)$, I assumed that $f_{\phi}$ is a deterministic function. However, to put it into the KL divergence, it would need to be a distribution, so the type signatures do not match. Perhaps the authors mean the Dirac delta distribution on the unique output? (I did not check whether that case then makes sense)

**3. Minor Weaknesses (Addressing those won't change my score):**
- Footnotes should be placed after punctuation marks, not before
- Sometimes, some clarity is missing. E.g., I needed a bit of time to figure out that $X$ denotes *input pairs* and $Y$ denotes *choices* in Section 3.2. I first thought it would be inputs and rewards. I'd recommend clarifying this. The confusion arose, e.g., because you denoted $Y$ as the "reward model output". (Besides, it's not even the preference model output but instead the true preference of the gold RM! Clarity on this might have prevented the wrong Markov chain I mention above.)
- I recommend the notation $I(X; Y)$ over $I(X, Y)$, since the latter could be confused with the entropy of the joint variable $(X, Y)$.

**Questions:**

See above.

**Limitations:**

Yes, though my impression was that the broader impacts and limitations should have been explained within the 9 page page-limit, though I am not entirely sure. In the submitted paper, both are on page 10. Maybe it's worth clarifying, should the paper be accepted, whether these sections should be within 10 pages or whether they're then allowed to even be on *page 11*.

---

> ### Author Rebuttal · Authors · 2024-08-06
>
> Thank you for acknowledging our strong empirical results. We will address each of your comments below and also in our revised manuscript.
>
> ---
>
> > **W1:** Some typos and missing assumptions in variational lower bound derivation.
> >
>
> **RW1:** Thank you for carefully checking our derivation process and pointing out the typos and missing assumptions. Based on your comments, we will modify the variational lower bound derivation process as follows:
>
> - First, we will correct the typo in the lower bound of  $I(Y; S)$ to $\mathbb{E} _{(x,y)} \left[ \int p _{\phi}(s \mid x) \log q _{\psi}(y \mid s)  ds \right]$ in the revised version.
>
> - Second, we will add the assumption about the hidden state distribution for Equation (29) in Appendix A of our paper:
> “*The distribution of the hidden state $p_{\phi}(s \mid x)$ follows a Gaussian distribution with the mean and variance determined by the output of the encoder* $f_{\phi}(\cdot)$”
>
> - Third, we will remove the proof of $I(X; S \mid Y) \leq I(X; S)$  in Appendix A of our paper and directly use it as a well-known inequality.
>
> It is worth noting that although there are some local issues in our variational lower bound derivation, such as typographical errors and missing assumptions, **they are primarily located in Appendix A and do not affect the final derivation results.** Therefore, **they barely impact other parts of the paper, and our overall work remains valid.**
>
> > **W2:** Question about **$KL[f_{\phi}(x_w), r(S)]$.**
> >
>
> **RW2:** Thanks for pointing this out. Based on the hidden state distribution assumption in RW1, this term should be corrected to $KL[p_{\phi}(s \mid x), r(s)]$. In the revised version, we will double-check our paper to avoid similar errors.
>
> > **W3:** Some unclear notations.
> >
>
> **RW3:** Thanks for your careful suggestion. In the revised version, we will correct the position of the footnotes, further clarify the definition of $X$ and $Y$, and replace $I(X,Y)$ by $I(X;Y)$.

---

> > ### Comment · Reviewer_YUjH · 2024-08-08
> >
> > Thank you for the rebuttal!
> >
> > > Second, we will add the assumption about the hidden state distribution for Equation (29) in Appendix A of our paper: “The distribution of the hidden state follows a Gaussian distribution with the mean and variance determined by the output of the encoder”
> >
> > - Could you write down in formulas the relationship between $f_{\phi}$ and $p_{\phi}$? Including the precise formula for mean and variance?
> > - As I said, assuming a Dirac distribution, I could make sense of the first half of Equation (29). Could you provide a full derivation based on your Gaussian assumption?
> >
> > > Third, we will remove the proof of $I(X;S \mid Y) \leq I(X;S)$ in Appendix A of our paper and directly use it as a well-known inequality.
> >
> > I actually recommend (and also stated so) to directly start with $I(X;S)$ in the objective in Equation (3) since this is closer to what you actually optimize and the standard in the literature.

---

> ### Author Response · Authors · 2024-08-11
> **Response to Reviewer YUjH [Part I]**
>
> > **Q1:** The relationship between $f_{\phi}$ and $p_{\phi}$.
> >
>
> **Response:** By assuming  latent representation distribution $p_{\phi}(\boldsymbol s| \boldsymbol x)$ follows a multivariate Gaussian with a diagonal covariance structure, whose mean and covariance are determined by the output of the encoder $f_{\phi}(\boldsymbol x)$, the relationship between $f_{\phi}(\boldsymbol x)$ and $p_{\phi}(\boldsymbol s|\boldsymbol x)$ can be formulated as follows:
>
> $p_{\phi}(\boldsymbol s \mid\boldsymbol x) = \mathcal{N}(\boldsymbol s \mid f_{\phi}^{\boldsymbol \mu}(\boldsymbol x), f_{\phi}^{\boldsymbol \sigma}(\boldsymbol x))=\frac{1}{\sqrt{(2\pi)^k |\boldsymbol \Sigma|}} \exp\left( -\frac{1}{2} (\boldsymbol s - f_{\phi}^{\boldsymbol \mu}(\boldsymbol x))^\top \boldsymbol \Sigma^{-1} (\boldsymbol x - f_{\phi}^{\boldsymbol \mu}(\boldsymbol x)) \right)$,
>
> where $\boldsymbol x$ is the network input, $\boldsymbol s$ is the latent representation, $f_{\phi}^{\boldsymbol \mu}(\boldsymbol x)$ is the mean of $\boldsymbol s$, and $\boldsymbol \Sigma$ is the diagonal covariance matrix determined by $f_{\phi}^{\boldsymbol \sigma}(\boldsymbol x)$. **Specifically, the encoder $f_{\phi}(\boldsymbol x)$  generates two outputs: $f_{\phi}^{\boldsymbol \mu}(\boldsymbol x)$ and $f_{\phi}^{\boldsymbol \sigma}(\boldsymbol x)$. The first output, $f_{\phi}^{\boldsymbol \mu}(\boldsymbol x)$, represents the $K$-dimensional mean of the latent representation $\boldsymbol s$. The second output, $f_{\phi}^{\boldsymbol \sigma}(\boldsymbol x)$ is squared to form the diagonal elements of the $K \times K$ diagonal covariance matrix $\boldsymbol \Sigma$.**
>
> In the revised version, We will further clarify the relationship between $f_{\phi}$ and $p_{\phi}$.

---

> ### Author Response · Authors · 2024-08-11
> **Response to Reviewer YUjH [Part II]**
>
> > **Q2:** Full derivation of Equation (29) based on Gaussian assumption.
>
> **Response:** The full derivation of Equation (29) in our paper is as follows:
>
> Let $\boldsymbol X$, $\boldsymbol  S$,  and $Y$ denote the random variable of reward model input,  latent representation, and human preference ranking respectively. The variational lower bound of our IB objective can be formulated as follows:
>
> $J(\boldsymbol{\theta})=I(\boldsymbol S;Y)-\beta I(\boldsymbol X;\boldsymbol S) \geq \mathbb{E} _{(\boldsymbol x,y)}\left[\int p _\phi(\boldsymbol s|\boldsymbol x) \log q _\psi(y | \boldsymbol s) d\boldsymbol s \right] - \beta\ \mathbb{E} _{\boldsymbol x}\left[KL(p _{\phi}(\boldsymbol S|\boldsymbol x)||r(\boldsymbol S))\right]=L$,
>
> where $r(\boldsymbol s)=\mathcal{N}(\boldsymbol{s};\mathbf{0},\mathbf{I})$ is the variational approximation of the marginal distribution $p(\boldsymbol s)$. Notably, $p_{\phi}(\boldsymbol{s}|\boldsymbol{x})$ is modeled as a multivariate Gaussian with a diagonal covariance structure, where the mean and covariance are both determined by the output of the encoder  $f_{\phi}(\boldsymbol{x})$, specifically $f_{\phi}^{\boldsymbol{\mu}}(\boldsymbol{x})$ for the mean and $f_{\phi}^{\boldsymbol{\sigma}}(\boldsymbol{x})$ for the covariance; please see the response to Q1 for their detailed relationship. Then, given a latent representation $\boldsymbol s$ drawn from $p_{\phi}(\boldsymbol s|\boldsymbol x)$, the decoder $g_{\psi}(\boldsymbol s)$ estimates the human preference ranking $y$ based on the distribution $q_{\psi}(y|\boldsymbol s)$.
>
> By estimating the expectation on $(\boldsymbol x, y)$ using the sample estimate based on the preference dataset  $\mathcal{D}=[\boldsymbol x_n,y_n] _ {n=1}^N$, where $\boldsymbol x_{n}$ comprises a human-chosen sample $\boldsymbol x_{n}^w$ and a human-rejected sample $\boldsymbol x_{n}^l$, with $y_n$ representing the corresponding human preference ranking, the variational lower bound of our IB objective can be approximated as follows:
>
> $L \approx \frac{1}{N} \sum_{n=1}^{N} \left[ \int  p_{\phi}(\boldsymbol s|\boldsymbol x_n) \log q_{\psi}(y_n|\boldsymbol s)d\mathbf s - \beta \ KL(p_{\phi}(\boldsymbol S|\boldsymbol x_n) ||r(\boldsymbol S)) \right].$
>
> Based on the Gaussian distribution assumption on $p_{\phi}(\boldsymbol s|\boldsymbol x)$, we can use the reparameterization trick to write $p(\boldsymbol s|\boldsymbol x)d\boldsymbol s = p(\boldsymbol  \epsilon)d\boldsymbol  \epsilon$, where $\boldsymbol \epsilon$ is an auxiliary Gaussian random variable with independent marginal $p(\boldsymbol \epsilon)$. In this way,  $\boldsymbol s$ can be expressed by a deterministic function $\boldsymbol s = h_{\phi}(\boldsymbol x,\boldsymbol \epsilon)=f _ {\phi}^{\boldsymbol \mu}(\boldsymbol x)+ f _ {\phi}^{\boldsymbol \sigma}(\boldsymbol x)\boldsymbol \epsilon$.
>  Hence, we can get the following objective function:
>
> $L \approx \frac{1}{N} \sum _ {n=1}^{N} \left[ \mathbb{E} _ {\boldsymbol \epsilon \sim p(\boldsymbol \epsilon)} \left[\log q _ {\psi}(y _ n | h _ {\phi}(\boldsymbol x _ n, \boldsymbol \epsilon)) \right] - \beta \ \text{KL} \left[ p _ {\phi}(\boldsymbol S|\boldsymbol x_n), r(\boldsymbol S) \right]\right]$
>
> In our experiments, we further employ a sample estimate to determine  $\mathbb{E} _ {\boldsymbol \epsilon \sim p _ (\boldsymbol \epsilon)} \left[\log q_{\psi}(y_n | h_{\phi}(\boldsymbol x_n, \boldsymbol \epsilon)) \right]$, by sampling a $\boldsymbol \epsilon$ from $p(\boldsymbol \epsilon)$, balancing computational complexity. Thus our objective can be estimated as follows:
>
> $L \approx \frac{1}{N} \sum_{n=1}^{N}  \left[  \log q_{\psi}(y_n | h_{\phi}(\boldsymbol x_n, \boldsymbol \epsilon)) - \beta \ \text{KL} \left[ p_{\phi}(\boldsymbol S|\boldsymbol x_n), r(\boldsymbol S) \right]\right]$ .
>
> **Due to word count limitations, the derivation continues in the next block, i.e., Response to Reviewer YUjH [Part III].**

---

> ### Author Response · Authors · 2024-08-11
> **Response to Reviewer YUjH [Part III]**
>
> **Continuing from the previous block, i.e., Response to Reviewer YUjH [Part II], we now complete our derivation.**
>
> As established in the previous block, we have derived that our objective can be estimated as follows:
>
> $L \approx \frac{1}{N} \sum_{n=1}^{N}  \left[  \log q_{\psi}(y_n | h_{\phi}(\boldsymbol x_n, \boldsymbol \epsilon)) - \beta \ \text{KL} \left[ p_{\phi}(\boldsymbol S|\boldsymbol x_n), r(\boldsymbol S) \right]\right]$ .
>
> According to the Bradley-Terry Model, the human preference distribution $p(y_n)$ can be formulated as:
>
> $p(y_n) = p(\boldsymbol x_{n}^w \succ \boldsymbol x_{n}^l)= \sigma(r(\boldsymbol x_{n}^w)-r(\boldsymbol x_{n}^l))$,
>
> where $\sigma(\cdot)$ is the logistic function, and $r(\cdot)$ is the reward model. Notably, in this work, reward model  $r(\cdot)$ consists of the previously mentioned encoder  $f_{\phi}(\cdot)$ and decoder $g_{\psi}(\cdot)$  and can be expressed as follows:
>
> $r(\boldsymbol x) = g_{\psi}(h_{\phi}(\boldsymbol x_n, \boldsymbol \epsilon))= g_{\psi}(f_{\phi}^{\boldsymbol \mu}(\boldsymbol x)+ f_{\phi}^{\boldsymbol \sigma}(\boldsymbol x)\boldsymbol \epsilon)$.
>
> Combining the two equations, we obtain:
>
> $\log q_{\psi}(y_n | h_{\phi}(\boldsymbol x_n, \boldsymbol \epsilon)) = \text{log}\ \sigma(g_{\psi}(h_{\phi}(\boldsymbol x_n^{w}, \boldsymbol \epsilon)) - g_{\psi}(h_{\phi}(\boldsymbol x_n^{l}, \boldsymbol \epsilon)))$ .
>
> Now, our estimation of the objective becomes:
>
> $L \approx \frac{1}{N} \sum_{n=1}^{N}  \left[ \text{log}\ \sigma(g_{\psi}(h_{\phi}(\boldsymbol x_n^{w}, \boldsymbol \epsilon)) - g_{\psi}(h_{\phi}(\boldsymbol x_n^{l}, \boldsymbol \epsilon))) - \beta \ \text{KL} \left[ p_{\phi}(\boldsymbol S|\boldsymbol x_n^w), r(\boldsymbol S) \right] - \beta \ \text{KL} \left[ p_{\phi}(\boldsymbol S|\boldsymbol x_n^l), r(\boldsymbol S) \right]\right]$ ,
>
> in which $\text{KL} \left[ p_{\phi}(\boldsymbol S|\boldsymbol x_n), r(\boldsymbol S) \right]$ is replaced by $\text{KL} \left[ p_{\phi}(\boldsymbol S|\boldsymbol x_n^w), r(\boldsymbol S) \right] + \text{KL} \left[ p_{\phi}(\boldsymbol S|\boldsymbol x_n^l), r(\boldsymbol S) \right]$.
>
> Recalling that $h_{\phi}(\boldsymbol x,\boldsymbol \epsilon)=f_{\phi}^{\boldsymbol \mu}(\boldsymbol x)+ f_{\phi}^{\boldsymbol \sigma}(\boldsymbol x)\boldsymbol \epsilon$, we abbreviate $g_{\psi}(h_{\phi}(\cdot, \boldsymbol \epsilon)))$ as $g_{\boldsymbol \psi} \circ f_{\boldsymbol \phi} (\cdot)$  for clarity and ease of understanding, leading to the final objective in our paper:
>
> $L \approx \frac{1}{N} \sum_{n=1}^{N}  \left[ \text{log}\ \sigma(g_{\boldsymbol \psi} \circ f_{\boldsymbol \phi}(\boldsymbol x_n^{w}) - g_{\boldsymbol \psi} \circ f_{\boldsymbol \phi}(\boldsymbol x_n^{l})) - \beta \ \text{KL} \left[ p_{\phi}(\boldsymbol S|\boldsymbol x_n^w), r(\boldsymbol S) \right] - \beta \ \text{KL} \left[ p_{\phi}(\boldsymbol S|\boldsymbol x_n^l), r(\boldsymbol S) \right]\right]$,
>
> where $\sigma(\cdot)$ is the logistic function.
>
> In the revised version, we will include this detailed derivation in the Appendix A

---

> > ### Author Response · Authors · 2024-08-11
> > **Response to Reviewer YUjH [Part IV]**
> >
> > > **Q3:** Suggestion about directly starting with  $I(X;S)$ instead of $I(X;S|Y)$
> > >
> >
> > **Response:** Thank you for your valuable suggestion. In the revised version, we will directly start by minimizing $I(X;S)$ and use this term consistently throughout the paper for greater clarity.

---

> > > ### Comment · Reviewer_YUjH · 2024-08-12
> > >
> > > Thank you for the response!
> > >
> > > I think I am still confused about some aspects of this derivation, but I also think providing this derivation already provides significant value to move the discussion forward.
> > >
> > > > Based on the Gaussian distribution assumption on $p_{\phi}(\boldsymbol s|\boldsymbol x)$, we can use the reparameterization trick to write $p(\boldsymbol s|\boldsymbol x)d\boldsymbol s = p(\boldsymbol  \epsilon)d\boldsymbol  \epsilon$, where $\epsilon$
> > >  is an auxiliary Gaussian random variable with independent marginal $p(\epsilon)$.
> > >
> > > Thank you, this seems like a really important part of the derivation that I'm happy to see more details on.
> > > To clarify, do you sample one $\epsilon$ for the whole pair $x^w, x^l$, or do you sample separate $\epsilon^w, \epsilon^l$ for the two parts?
> > >
> > > Furthermore, could you explain why $\epsilon$ disappears in the final part of the derivation?

---

> > > > ### Author Response · Authors · 2024-08-12
> > > > **Response to Reviewer YUjH**
> > > >
> > > > > **Q1:** To clarify, do you sample one $\boldsymbol \epsilon$ for the whole pair $\boldsymbol x^{w}$, $\boldsymbol x^{l}$, or do you sample separate $\boldsymbol \epsilon^w$, $\boldsymbol \epsilon^l$ for the two parts?
> > > >
> > > > **Response:** For each input sample, either $\boldsymbol x_n^w$ or $\boldsymbol x_n^l$, where $n$ ranges from 1 to $N$, we independently sample an $\boldsymbol \epsilon$ from the standard multivariate Gaussian distribution, denoted as $p(\boldsymbol \epsilon) = \mathcal{N}(\mathbf{0}, \mathbf{I})$, to generate the latent representation. Specifically, for $\boldsymbol x_n^w$, the corresponding latent representation is given by $\boldsymbol s_n^w = h_{\phi}(\boldsymbol x_n^w, \boldsymbol \epsilon_n^w)$, where $\boldsymbol \epsilon_n^w \sim \mathcal{N}(\mathbf{0}, \mathbf{I})$. Similarly, for $\boldsymbol x_n^l$, the corresponding latent representation is $\boldsymbol s_n^l = h_{\phi}(\boldsymbol x_n^l, \boldsymbol \epsilon_n^l)$, where $\boldsymbol \epsilon_n^l \sim \mathcal{N}(\mathbf{0}, \mathbf{I})$. We will distinguish between $\boldsymbol \epsilon_n^w$ and $\boldsymbol \epsilon_n^l$ in the revised version to prevent any confusion. In addition, this approach, where $\boldsymbol \epsilon_n^w$ or $\boldsymbol \epsilon_n^l$ is independently sampled for each corresponding input sample, is consistently applied across all our implementations.
> > > >
> > > >
> > > > > **Q2:** Furthermore, could you explain why $\epsilon$ disappears in the final part of the derivation?
> > > >
> > > > **Response:**  We would like to clarify that $\boldsymbol \epsilon$ does not cancel out in our derivation. For simplicity, we just use $g_{\boldsymbol \psi} \circ f_{\boldsymbol \phi} (\cdot)$ as an abbreviation for $g_{\psi}(h_{\phi}(\cdot, \boldsymbol \epsilon))$. Based on your suggestion, we will eliminate this abbreviation and directly utilize the intermediate product as our final objective:
> > > > $L \approx \frac{1}{N} \sum_{n=1}^{N} \left[ \log \sigma \left( g_{\psi}(h_{\phi}(\boldsymbol x_n^{w}, \boldsymbol \epsilon_n^{w})) - g_{\psi}(h_{\phi}(\boldsymbol x_n^{l}, \boldsymbol \epsilon_n^{l})) \right) - \beta\ \text{KL} \left[ p_{\phi}(\boldsymbol S|\boldsymbol x_n^w), r(\boldsymbol S) \right] - \beta\ \text{KL} \left[ p_{\phi}(\boldsymbol S|\boldsymbol x_n^l), r(\boldsymbol S) \right] \right]$,
> > > >
> > > > where $\boldsymbol \epsilon_n^{w}$ and $\boldsymbol \epsilon_n^{l}$ are independently sampled from $\mathcal{N}(\mathbf{0}, \mathbf{I})$ for each input sample.

---

> > > > > ### Comment · Reviewer_YUjH · 2024-08-12
> > > > >
> > > > > Thank you for the further details!
> > > > >
> > > > > Some last clarification questions: When you write $\mathbf{S}$, is it true that sometimes you mean a tuple $(\mathbf{S}^w, \mathbf{S}^l)$, and sometimes a single vector (depending where you are in the derivation)? Likewise, is $r(S)$ sometimes an independent product distribution over tuples, and sometimes a single Gaussian distribution? Likewise, is $p_{\phi}(s \mid x)$ sometimes a product $p_{\phi}(s^w \mid x^w) \cdot p_{\phi}(s^l \mid x^l)$?
> > > > >
> > > > > If so, then I think it will help some readers enormously to be clearer by distinguishing these notations.

---

> ### Author Response · Authors · 2024-08-13
> **Response to Reviewer YUjH**
>
> > **Q1:** Some clarification questions about $\boldsymbol S$.
>
> **Response:** Thank you for your valuable suggestion.  You are indeed correct, and we recognize that the notation for $\boldsymbol S$ has caused some confusion. In the revised version of our manuscript, we intend to make the following clarifications to address the points you raised:
>
> 1. We will clarify that the notation $\boldsymbol S$ represents the tuple  $(\boldsymbol S^w, \boldsymbol S^l)$, where $\boldsymbol S^w$ and $\boldsymbol S^l$ denote the latent representation corresponding to the accepted and rejected samples, respectively.
> 2. We will replace $p_{\phi}(\boldsymbol s|\boldsymbol x_n^w)$ with $p_{\phi}(\boldsymbol s^w|\boldsymbol x_n^w)$, and similarly, $p_{\phi}(\boldsymbol s|\boldsymbol x_n^l)$  will be replaced with $p_{\phi}(\boldsymbol s^l|\boldsymbol x_n^l)$.
> 3. We will clarify that  $p_{\phi}(\boldsymbol s|\boldsymbol x_n)$ is equivalent to $p_{\phi}(\boldsymbol s^w|\boldsymbol x_n^w) \cdot p_{\phi}(\boldsymbol s^l|\boldsymbol x_n^l) .$
> 4. We will further clarify that $r(\boldsymbol S)$  is the independent product distribution over the tuple $(\boldsymbol S^w, \boldsymbol S^l)$, i.e., for an instance $(\boldsymbol s^w, \boldsymbol s^l)$, $r(\boldsymbol s)= r(\boldsymbol s^w) \cdot r(\boldsymbol s^l)$, where  $r(\boldsymbol s^w)$ and $r(\boldsymbol s^l)$ each represent a single standard Gaussian distribution $\mathcal{N}(\mathbf{0}, \mathbf{I})$.
>
> Furthermore, we will thoroughly review our entire paper to prevent similar questions. We appreciate your valuable suggestion, which has significantly helped us enhance the clarity and precision of our derivations.

---

> > ### Comment · Reviewer_YUjH · 2024-08-13
> > **Summary of the discussion**
> >
> > Thank you for answering my last clarification questions!
> >
> > Here's a summary of the discussion (e.g., for the area chair):
> >
> > In the discussion, the authors provided a correct derivation of their optimization objective. Crucially, the discussion **revealed that the authors sample Gaussian noise via a reparameterization trick**. This was entirely omitted from the original submission, and thus the optimization objective in the discussion looks a bit different from the one found in the original submission. Importantly, **this is not just a detail in the derivation, but impacts the concrete implementation of the method**.
> >
> > I am increasing my score to 5 (borderline accept), which comes with some trust that:
> >
> > - The authors are able to use the derivations in the discussion as a basis to write up a coherent derivation in the paper;
> > - It is actually true that the authors have used the reparameterization trick in their implementation as revealed in this discussion.
> >
> > In my opinion, it depends on the area chair's trust in these two points (together with the other reviewer's discussions) whether the paper can be accepted.

---

> ### Author Response · Authors · 2024-08-13
> **Confirmation of the implementation of the reparameterization trick**
>
> Reviewer YUjH,
>
> We sincerely appreciate your recognition of our work and your timely responses during the rebuttal process. Additionally, **the reparameterization trick has indeed been used in our implementation**. **The related evidence can be found in our submitted code** from Line 331 to Line 338 in `InfoRM_code/utils/model/reward_model.py` for your kind reference.
>
> Best regards.

---

### Official Review · Reviewer_XuFQ · 2024-06-25

**Soundness:** 3
**Presentation:** 2
**Contribution:** 3
**Rating:** 6
**Confidence:** 4

**Summary:**

This paper presents a novel way to train reward models form human preferences using an information bottleneck architecture and training method. They provide a derivation of how to train a reward model with an information bottleneck, and then produce empirical evidence of improved performance when using the InfoRM vs several baselines in the AlpacaFarm instruction-following setting. Compared with a single reward model and mean-ensemble, their results show their method has better performance and a better reward-KL frontier. The authors also introduce a method, Cluster Separation Index (CSI), which utilises the latent space of the IB to detect when overoptimisation of the reward model may be happening. They provide qualitative evidence that this CSI does somewhat correspond to overoptimisation occurring.

**Strengths:**

The issue of overoptimisation, and improving RLHF in general, is one that is very important to the community, and so this paper's focus on that problem is beneficial.

The paper's utilisation of the IB approach to reward modelling is novel (to my knowledge), and demonstrates encouraging performance improvements.

The quality of the work is reasonable. The method is demonstrated to work well vs various baselines empirically, and several evaluations are used to back up this conclusion.

The dual use of the method as both increasing RLHF performance and producing an interpretable latent space for detecting overoptimisation is an added benefit of the methodology.

**Weaknesses:**

## Bigger Points

### Insufficient and imperfect comparison to baselines

In general, the comparison to baselines in all the settings you consider isn't sufficient to justify the method is outperforming the existing SOTA. You don't compare against WCO or UWO from https://arxiv.org/abs/2310.02743, or WARM from https://arxiv.org/abs/2401.12187 (or a variant of it that just uses the ensemble methods), or https://arxiv.org/abs/2403.05171. Some of these are contemporary, but WCO, UWO, WARM aren't given they're all at least 4 months old from the time of submission, and so comparing against them is important.

Further, there is a lack of clarity about how hyperparameters were chosen for each method (e.g. KL, learning rate), including baselines, which makes it unclear how much the hyperparameters were optimised for your method vs the baselines. Picking 1 learning rate for all methods means that if that learning rate is optimal for your method, it's not necessarily optimal for other methods, and which is an unfair advantage to your method. Having a systematic way of choosing hyperparameters for all methods would make the comparison to the baselines much more compelling. This is particularly important as your method introduces two new hyperparameters on top of some of the baselines.

One of the ppo training choices is suboptimal - "The critic model was initialized with the weight of the SFT model" - it should be set to the reward model, as that's been shown in previous work to improve performance (https://arxiv.org/abs/2009.01325). This could affect all methods, or it could affect different methods differently, so it would be useful to correct this choice in the evaluations of your method.

Finally, while you do compare to some baselines in the simulated setting, you only compare to very weak baselines in the realistic setting, meaning it's impossible to know whether your method is actually outperforming existing work in this setting.

### Limited Evaluation
You only perform experiments on one datasets, with a single policy size and a small range of reward model sizes. It would be beneficial to perform experiments on another dataset (for example TL;DR summarisation or anthropic-HH). Additional sizes would also be beneficial but are less important that another dataset to demonstrate the generality of your method.

Further, it seems that all results are only for a single seed. Running multiple seeds is important in this setting, as performance can vary substantially between seeds.

### Non-neutral language describing their own method and contributions.
Throughout the paper, the language describing your method and contributions is not sufficiently neutral, which makes the paper harder to read and less clear. It would be better if this over-enthusiastic language could be toned down through the paper. Some examples:
* 52-53 "Which cleverly addresses" - you don't need "cleverly" here
* 66 "We surprisingly discover" - no need for "surprisingly"
* 68 "meticulously" - as above
* 73 "significantly *mitigates* the risk" - this is specifically egrerious as "significantly" is often taken to mean statistically significantly, but you perform no statistical tests throughout the paper.
* 74 "*marked* improvement in RLHF performance"

## Smaller Points

* There's a typo in the legend of figure 5: sandard -> standard.

### Unclear motivation for your method

Throughout the paper you argue that misgeneralisation is an important cause of overoptimisation, that existing methods don't tackle this but that your method does. Due to the reliance on this argument to motivate your method, it would be useful to see a clearer definition of what you mean by misgeneralisation, and how existing methods don't tackle it. In my mind, increasing model size or training an ensemble are both ways of producing models that generalise better and are more robust (not just when training RMs), so claiming that these methods don't tackle misgeneralisation needs further clarity and justification.

### Lack of quantitative evaluation of CSI

The CSI method and interpretability of the IB space are promising additional benefits of your method. However, it would be useful to see quantitative results of how well using CSI as an early stopping method actually works vs other baselines (e.g. no early stopping, or early stopping on other candidate metrics), to get a sense of whether this method would actually be useful or not in practice.

## Summary

Overall, I think the proposed method has some promise, but the existing empirical results aren't sufficient to demonstrate it's effectiveness vs the current literature, and the clarity and motivation of the paper is unclear and limited by overly enthusiastic writing. I'm currently reccomending a reject. If the issues with comparisons to baselines were fully addressed and the method still performed better than existing approaches I would raise my score to a weak accept, and additionally if the language of the paper was adjusted, better evaluation of CSI was provided and evaluation with multiple seeds and over additional datasets was provided then I would raise my score to an accept. However, as it stands I don't believe the paper is worthy of acceptance.

[EDIT]: I have raised my score to a 6. The new results show much more convincingly that this method is better than the baselines compared against.

**Questions:**

(Some of these are covered in the weaknesses section, but I will repeat them here).

* How did you select the hyperparameters for all your methods?
* Could you report reward model accuracy scores under standard training and your method, both In-distribution and out-of-distribution (e.g. to human preferences on the alpaca_eval test set). That would help with seeing whether your method is producing better RLHF performance because of better RM accuracy/generalisation, or due to another reason.

**Limitations:**

The authors discuss some of the limitations of their work, but they don't discuss any of those I broad up in the Weaknesses section, which I think are all major limitations of the work.

---

> ### Author Rebuttal · Authors · 2024-08-06
>
> Thank you for acknowledging the novelty, reasonability, and performance improvements of our method. We will address each of your comments below and also in our revised manuscript.
>
> ---
>
> > **W1:** Comparison with UWO or WARM.
>
> **RW1**: Thanks for your valuable comment. We would like to clarify that we have compared our method with Ensemble RM, i.e., UWO proposed in [1], in our simulated experiments. However, we did not include this comparison in our real-world experiments focusing on larger RMs due to computational constraints, as UWO requires loading multiple RMs during the RL process.
>
> Following your suggestion, **we further include WARM [2] in our real-world experiments.** The results, reported in Table 1 of the submitted PDF, demonstrate that **our InfoRM achieves better RLHF performance as compared with WARM.**
>
> Additionally, we would like to highlight that **our InfoRM is a fundamental and flexible framework that can easily integrate with other techniques to complement each other.** The results in Table 1 of the submitted PDF indicate that integrating InfoRM with WARM  further enhances RLHF performance.
>
> > **W2**: Systematic hyperparameters selection for compared methods.
>
> **RW2:** Thanks for your suggestion.  **For all methods in our real-world experiments, we directly use the recommended hyperparameters from the widely recognized technical reports** [3,4], i.e., a learning rate of 5e-7 and a KL penalty of 0.001 where applicable.
>
> To further address your concern, we report the performance of each compared method with different hyperparameter settings in Table 2 of the submitted PDF. The results show that **a learning rate of 5e-7 and a KL penalty of 0.001 are indeed the optimal settings for all methods**, validating the fairness and reliability of our experiments.
>
> > **W3:** The choice of initializing critic model from SFT model.
>
> **RW3:** Thanks for your careful feedback. We would like to clarify that while we modify the network structure of the reward model, we do not alter the structure of the critic model to control the variables. **Due to these structural differences, we cannot initialize the critic model from our modified reward model**. To ensure fair comparisons, all critic models in our experiments are initialized from the SFT model.
>
> Additionally, we would like to kindly argue that **initializing the critic model using the SFT model is also a viable option.** As stated in [4]: "Initializing the critic model with a reward or SFT model will converge to similar results."
>
> > **W4:** More datasets and reward/policy model sizes in our experiment.
>
> **RW4:** We would like to kindly argue that **our experiments are not limited to one dataset, one policy model, and a small range of reward model sizes.** We clarify our experimental settings in the table below:
>
> ||Datasets|Reward Model Size|Policy Model Size|
> |---|---|---|---|
> |Simulated Experiments|AlpacaFarm| 70M, 410M, and 1.4B|1.4B|
> |Real-world Experiments|Anthropic-HH|7B|7B|
>
> Following your suggestion, **we also add the TL;DR summarization dataset to our real experiments**. The results, reported in Table 1 of the submitted PDF, **show that our InfoRM outperforms the compared methods on this task as well.**
>
> > **W5:**  Running multiple seeds is important.
>
> **RW5:** Following your suggestion, we conduct our real experiments with a new seed (100) and report the results on the AlpacaFarm dataset in the table below. Due to time and resource constraints, we will include the results from more seeds in the revised version to ensure a robust evaluation.
> |Model|Opponent|Win|Tie|Lose|
> |---|---|---|---|---|
> ||Standard RM|53.1|21.3|25.6|
> |InfoRM|Standard RM w/KL|42.3|28.6|29.1|
> ||WARM|38.3|31.5|30.2|
>
> > **W6:** More neutral descriptions.
>
> **RW6:**  Thank you for your suggestion.  We will carefully review the entire paper and revise any descriptions that lack neutrality to enhance the readability of our paper.
>
> > **W7:** Why existing methods of increasing model size or using ensemble models cannot effectively solve misgeneralization？
>
> **RW7:** The underlying principle behind these existing methods is to ***implicitly* remove spurious features by increasing the reward model's capability**, which fails to directly address this issue and results in an inefficient solution.
>
> In contrast, **our method *explicitly* identifies and eliminates spurious features more efficiently**. Specifically, our InfoRM achieves this by maximizing the utility of the latent representation for reward prediction while minimizing irrelevant human preferences. Our experimental results show InfoRM's superiority over existing methods with the same model size.
>
> > **W8:** Effectiveness of our CSI as an early stopping method.
>
> **RW8:** Based on your suggestion, we report the comparison results of our InfoRM with and without early stopping in Table 3 of the submitted PDF. As shown, **the early stopping strategy according to our CSI metric indeed enhances RLHF performance**, particularly on the Anthropic-Harmless dataset.
>
> > **Q1:** The accuracy comparison between our InfoRM and Standard RM.
>
> **RQ1:** Thanks for your valuable feedback. To address your concern, we report the accuracy of InfoRM and Standard RM on in-distribution reward model benchmarks (Anthropic-Helpful and Anthropic-Harmless) and out-of-distribution reward model benchmarks (AlpacaEval and Truthful QA) in Table 4 of the submitted PDF. The results demonstrate that **our InfoRM achieves better RM accuracy and generalization, leading to improved RLHF performance**.
>
> [1] Coste, Thomas, et al. "Reward Model Ensembles Help Mitigate Overoptimization." ICLR 2024.
>
> [2] Rame, Alexandre, et al. "WARM: On the Benefits of Weight Averaged Reward Models." ICML 2024.
>
> [3] Bai, Yuntao, et al. "Training a helpful and harmless assistant with reinforcement learning from human feedback." *arXiv preprint* (2022).
>
> [4] Zheng, Rui, et al. "Secrets of RLHF in large language models part i: PPO." *arXiv preprint* (2023).

---

> > ### Comment · Reviewer_XuFQ · 2024-08-10
> > **Response**
> >
> > Thank you for your detailed response, and all the new results.
> >
> > > We would like to clarify that we have compared our method with Ensemble RM, i.e., UWO.
> >
> > To clarify, when you write "Ensemble RM" you mean UWO? It would be better if this was made clearer in the paper, as I would take "Ensemble RM" to just describe using a mean ensemble rather than UWO. Given you are using UWO, how do you select the variance coefficient hyperparameter?
> >
> > It would also be beneficial to compare to WCO and mean ensemble from Coste et al. for a more thorough set of baselines.
> >
> > > the early stopping strategy according to our CSI metric indeed enhances RLHF performance
> >
> > How exactly do you do early stopping with the CSI metric? Could you describe the algorithm here.
> >
> > Overall, the new results present a much more convincing story of the methods improved performance compared to the baselines. I will raise my score to a 6 (weak accept). I would still like to see comparisons with ensemble methods in the realistic setting, even if they are more computationally expensive. If these were presented and InfoRM outperformed them I would raise my score to a 7.

---

> ### Author Response · Authors · 2024-08-14
> **Response to Reviewer XuFQ [Part I]**
>
> > **Q1:** To clarify, when you write "Ensemble RM" you mean UWO? It would be better if this was made clearer in the paper, as I would take "Ensemble RM" to just describe using a mean ensemble rather than UWO. Given you are using UWO, how do you select the variance coefficient hyperparameter?
> >
>
> **RQ1:** Thank you for your valuable suggestion. In the revised version of our paper, we will use "Ensemble RM (UWO)" to refer to the UWO method. Additionally, for the two new baselines that we plan to add, namely Mean and WCO, we will use "Ensemble RM (Mean)" and "Ensemble RM (WCO)" to refer to them, respectively. The experimental results for the new baselines are reported in the response to Q2.
>
> Regarding **the selection of the variance coefficient hyperparameter for Ensemble RM (UWO)** in our simulated experiments, we directly use the recommended value (i.e.,$\lambda$ = 0.1) for PPO from [1], as our simulated experimental setup aligns closely with that in [1]. To ensure fairness and reliability, we report the simulated RLHF performance of Ensemble RM (UWO) with different variance coefficients in the table below. The results demonstrate that $\lambda$ = 0.1 is indeed the optimal choice in our simulated experiments.
>
> | Variance coefficient $\lambda$ | 0.05 | 0.1 | 0.5 | 1.0 |
> | --- | --- | --- | --- | --- |
> | Final gold score | 5.71 | 5.96 | 5.68 | 5.43 |
>
>
> > **Q2:** It would also be beneficial to compare to WCO and mean ensemble from Coste et al. for a more thorough set of baselines. &  I would still like to see comparisons with ensemble methods in the realistic setting
> >
>
> **RQ2:** Following your suggestion, we further include the Ensemble RM (UWO), Ensemble RM (WCO), and Ensemble RM (Mean) [1] in our real-world experiments. The comparison results are presented in the following table. To ensure fairness and reliability, we also report the performance of Ensemble RM (UWO) with different variance coefficients. The optimal settings, selected based on the highest win ratio, are highlighted in bold. The results show that **our InfoRM using a single RM achieves better RLHF performance compared to the ensemble methods**, and **integrating InfoRM with the ensemble methods further enhances RLHF performance**. We will also include the Ensemble RM (WCO) and Ensemble RM (Mean) in our simulated experiments in the revised version.
>
> | Model | Opponent |  |  | Anthropic Helpful |  |  | Anthropic Harmless |  |  | AlpacaFarm |  |  | TL;DR Summary |
> | --- | --- | --- | --- | --- | --- | --- | --- | --- | --- | --- | --- | --- | --- |
> |  |  | Win | Tie | Lose | Win | Tie | Lose | Win | Tie | Lose | Win | Tie | Lose |
> |  | Ensemble RM (WCO) | 47.2 | 35.5 | 17.3 | 52.1 | 39.9 | 8.0 | 39.8 | 35.1 | 25.1 | 63.9 | 23.4 | 12.7 |
> |  | Ensemble RM (Mean) | 48.7 | 32.9 | 18.4 | 54.0 | 35.9 | 10.1 | 41.7 | 38.3 | 20.0 | 65.3 | 24.7 | 10.0 |
> | InfoRM | Ensemble RM (UWO) ($\lambda$=0.5) | 46.9 | 31.8 | 21.3 | 50.5 | 33.5 | 16.0 | 38.2 | 35.9 | 25.9 | 62.9 | 27.5 | 9.6 |
> |  | **Ensemble RM (UWO) ($\lambda$=0.1)** | 43.1 | 33.1 | 23.8 | 49.3 | 34.8 | 15.9 | 37.3 | 37.8 | 24.9 | 61.4 | 28.1 | 10.5 |
> |  | Ensemble RM (UWO) ($\lambda$=0.05) | 43.5 | 33.6 | 22.9 | 50.1 | 35.1 | 14.8 | 37.8 | 35.3 | 26.9 | 61.6 | 29.3 | 9.1 |
> | InfoRM + Ensemble RM (UWO) ($\lambda$=0.1) | Ensemble RM (UWO) ($\lambda$=0.1) | 48.7 | 35.7 | 15.6 | 52.5 | 35.1 | 12.4 | 41.2 | 38.2 | 20.6 | 63.3 | 30.1 | 6.6 |
>
> [1] Coste, Thomas, et al. "Reward Model Ensembles Help Mitigate Overoptimization." ICLR 2024.

---

> ### Author Response · Authors · 2024-08-14
> **Response to Reviewer XuFQ [Part II]**
>
> > **Q3:** How exactly do you do early stopping with the CSI metric? Could you describe the algorithm here.
> >
>
> **RQ3:** We will **first provide the details of our previous early stopping validation experiments**, explaining how we use the CSI metric to select the stopping point during model training. **Following this, we will provide an automated early-stopping algorithm** based on our CSI metric for executing early stopping.
>
> **Early Stopping Validation Experimental Details**: In this experiment, we implemented early stopping by saving multiple checkpoints, visually inspecting their CSI values, and selecting the one before a significant increase in the CSI metric as the final checkpoint. This process is validated by our observations that overoptimization correlates with a significant increase in the CSI metric, making visual inspection effective, as demonstrated in Section 5 of our paper. However, we acknowledge that automating this process by quantifying CSI metric changes would be more cost-effective. Below, we provide an automated early-stopping algorithm based on the CSI metric.
>
> **Automated Early Stopping Algorithm Based on the CSI Metric**: The CSI-based early stopping algorithm is detailed as follows:
>
> 1. Set a maximum tolerable CSI change rate, $\epsilon_{\text{max}}$, which is empirically set to a relatively large value of 10. Let $C_t$ represent the CSI value at the $t$-th evaluation step. The change in CSI at this step is given by $\Delta_t = |C_t - C_{t-1}|$.
> 2. Calculate the ratio of the CSI change at the $t$-th evaluation step, $\Delta_t$, to the average change across all previous steps,  $\frac{1}{t-1} \sum_{i=1}^{t-1} \Delta_i$. This ratio is denoted as $\epsilon_t=\Delta_t / (\frac{1}{t-1} \sum_{i=1}^{t-1} \Delta_i)$.
> 3.  If $\epsilon_t > \epsilon_{\text{max}}$, trigger early stopping and exit the iteration. Otherwise, continue training.
>
> To facilitate understanding, we summarize this algorithm as follows:
>
> ---
>
> **Input:** Maximum tolerable CSI change rate $\epsilon_{\text{max}}$, initial CSI value $C_0$, maximum steps $T$
>
> **Initialize:** $C_{\text{prev}} \gets C_0$
>
> 1. **For** $t \gets 1$ to $T$ **do**:
>     1. Update model parameters.
>     2. $C_t \gets$ `evaluate_CSI(model)`
>     3. $\Delta_t \gets |C_t - C_{\text{prev}}|$
>     4. $\epsilon_t=\Delta_t / (\frac{1}{t-1} \sum_{i=1}^{t-1} \Delta_i)$
>     5. **If** $\epsilon_t > \epsilon_{\text{max}}$ **then**:
>         1. Trigger early stopping and exit loop.
>         2. **Break**
>     6. $C_{\text{prev}} \gets C_t$
>
> **Output:** Final model before early stopping.
>
> ---

---

> ### Author Response · Authors · 2024-08-14
> **Thanks for your response!**
>
> Dear Reviewer XuFQ,
>
> Thank you for your positive feedback and for raising your score. We appreciate your detailed review and valuable comments. We hope our latest responses further address your concerns.
>
> Best regards.

---

### Official Review · Reviewer_roDN · 2024-07-09

**Soundness:** 3
**Presentation:** 4
**Contribution:** 3
**Rating:** 6
**Confidence:** 3

**Summary:**

This paper proposes a variational information bottleneck (IB) objective for rewarding modeling in RLHF to mitigate the reward misgeneralization issue, which can cause overoptimization. The authors propose a variational information bottleneck objective to filter out irrelevant information and identify a correlation between overoptimization and outliers in the IB latent space. They also introduce the Cluster Separation Index (CSI) as an indicator for detecting reward overoptimization. Experimental results reveal the representation of IB latent space can be used to form an indicator of reward overoptimization.

**Strengths:**

The paper is well-motivated, addressing a critical issue in RLHF, and is well-written with clear explanations of the introduced concepts.
Mitigating the reward misgeneralization issue using IB seems to be a good idea. The introduction of CSI as an overoptimization detection tool is a valuable contribution, though it is only constrained to InfoRM.

**Weaknesses:**

Lack of reward model evaluations. Since the motivation of the paper is to mitigate the reward misgeneralization issue, there is no results to directly support such the claim as far as I am concerned.
It is not clear whether IB loss would affect the reward modeling abilities, such as accuracies and OOD generalization abilities.

**Questions:**

Does IB loss total address the misgeneralization issue? If not, when is IB inefficient? Could you provide the Upper bound of generalization error?
Using GPT-4 to identify overoptimization is promising. Nevertheless, it’d be nice to study how much it align with human identifications.
What is the architecture of RM encoder?

**Limitations:**

The additional computational overhead of training InfoRM is not discussed.

---

> ### Author Rebuttal · Authors · 2024-08-05
>
> Thank you for acknowledging the clarity of our paper and recognizing the potential of using IB to address reward misgeneralization in RLHF. We appreciate your positive feedback on the introduction of CSI as a valuable contribution. We will address each of your comments and concerns below and also in our revised manuscript.
>
> ---
>
> > **W1:** Direct results to support reward misgeneralization/ hacking mitigation.
>
> **RW1:**  Thanks for your feedback. Notably, there is no direct metric to assess reward misgeneralization/hacking. Currently, the only way to demonstrate reward misgeneralization/hacking is by observing the different trends of gold and proxy scores during the RLHF process in the simulated experiments, **which we have conducted in our paper and the results demonstrate the effectiveness of our method in reward misgeneralization/hacking mitigation**.
>
>  In addition, we also propose CSI as an auxiliary metric to measure reward hacking from the perspective of latent embeddings. Measuring hacking more effectively remains an open research question.
>
> > **W2:** Whether IB loss would affect the reward modeling abilities.
>
> **RW2:**  Thanks for your feedback. To address your concern, we report the accuracy of InfoRM and Standard RM on both in-distribution reward model benchmarks (Anthropic-Helpful and Anthropic-Harmless) and out-of-distribution reward model benchmarks (AlpacaEval and Truthful QA) in Table 4 of the submitted PDF. Our results show that **IB loss can also enhances reward modeling abilities**. Furthermore, the notable improvement in RLHF performance achieved by our InfoRM, as demonstrated in our paper, further substantiates this claim.
>
> > **Q1:** Does IB loss total address the misgeneralization issue? If not, when is IB inefficient?
>
> **RQ1:** Our InfoRM specifically addresses the issue of reward misgeneralization from an optimization perspective. While our method significantly reduces misgeneralization, we acknowledge that **completely addressing the issue is challenging due to the uncontrollable quality of preference datasets in practical applications**. Specifically, IB loss may be less effective in scenarios where the dataset quality is poor or highly variable. Under such conditions, the balance parameters of our method must be carefully adjusted to optimize the trade-off between accurate reward modeling and effective mitigation of reward misgeneralization. We will add these analysis in the revised version.
>
> > **Q2:** Could you provide the Upper bound of the generalization error?
>
> **RQ2:** The upper bound of the generalization error for our method is provided in Theorem 1 below, with the proof available in [1]. **Theorem 1 demonstrates that the mutual information between the latent representation and observations, as well as the latent space dimensionality, upper bound the expected generalization error of our InfoRM method.**
>
> ***Theorem 1:*** *Let $|S|$ be the cardinality of the latent representation space of InfoRM, $l(\cdot)$ be the loss function following sub-$\sigma$-Gaussian distribution, $X$ be the reward model input, $S$ be the latent representation of InfoRM, and $\Theta$ be the network parameters, we have the following upper bound for the expected generalization error of our InfoRM:*
>
> $$E[R(\Theta) - R_T(\Theta)] \leq \exp \left( -\frac{L}{2} \log \frac{1}{\eta} \right) \sqrt{\frac{2\sigma^2}{n} \log I(X,S)}\leq \exp \left( -\frac{L}{2} \log \frac{1}{\eta} \right) \sqrt{\frac{2\sigma^2}{n} \log |S|},$$
>
> *where  $L$, $\eta$, and $n$ are the effective number of layers causing information loss, a constant smaller than 1, and the sample size, respectively.  $R(\Theta) = \mathbb{E}_{X \sim D}[l(X, \Theta)]$ is the expected loss value given  $\Theta$  and  $R_T(\Theta) = \frac{1}{n} \sum _{i=1}^{n} l(X_i, \Theta)$ is a sample estimate of  $R(\Theta)$  from the training data.*
>
> > **Q3:**  How much GPT-4 align with human in reward hacking identifications?
>
> **RQ3:** We follow your valuable suggestion and **conduct a human evaluation to validate GPT-4 as the hacking annotator**. Specifically, we randomly sample 100 cases each from Anthropic Helpful, Anthropic Harmless, and AlpacaFarm. Then we engage two expert annotators proficient in alignment studies of LLMs and fluent in English. We ask them to evaluate these cases’ hacking phenomenon based on our pre-given descriptions of the hacking phenomena and the inter-annotator agreement rate is 96%. For cases where the annotators disagreed, we requested that both annotators reassess their evaluations to reach a consensus. The annotation serves as the reference to calculate the accuracy of the GPT-4-based  evaluator in reward hacking identification. **We find that the human-GPT agreement rate avengingly achieves a remarkable 96.7%, indicating the enhanced reliability of GPT-4 annotations in hacking detection.** We will include these results in the revised version.
>
> ||Anthropic Harmless|Anthropic Helpful|AlpacaFarm|
> |---|---|---|---|
> |human-GPT agreement|95%|98%|97%|
>
> > **Q4**: What is the architecture of the RM encoder?
>
> **RQ4:** In our experiments, the RM encoder is derived from the standard RM, with modification to the final layer.
>
> > **L1**: The additional computational overhead of training InfoRM.
>
> **RL1**: Thanks for your feedback. In fact, the primary consumption of time in RLHF occurs during the RL process. **Although the introduction of IB loss does indeed introduce some additional computational overhead for InfoRM training, its impact on the overall RLHF process is minimal.** To demonstrate this, we report the empirical estimates of the time costs for reward model training and RL process using Standard RM and our InfoRM in the table below.
>
> ||RM Training|RL Process|Overall|
> |---|---|---|---|
> |Standard RM| 0.35h | 9.00 h | 9.33 h |
> |InfoRM|0.55h| 9.00 h | 9.55 h |
>
>
> [1] Zhang, Sen, Jing Zhang, and Dacheng Tao. "Information-Theoretic Odometry Learning." IJCV 2022.

---

> > ### Comment · Reviewer_roDN · 2024-08-13
> >
> > Thanks to the authors' responses and additional experiments. My concerns are clarified. I lean to hold my previous decision.

---

> > > ### Author Response · Authors · 2024-08-14
> > > **Thanks for your response!**
> > >
> > > Dear Reviewer roDN,
> > >
> > > We sincerely appreciate your feedback regarding our efforts to address your concerns, and we would like to express our gratitude for your positive support.
> > >
> > > Best regards.

---

### Official Review · Reviewer_rGMr · 2024-07-10

**Soundness:** 3
**Presentation:** 2
**Contribution:** 3
**Rating:** 5
**Confidence:** 2

**Summary:**

The paper proposes a regularization method to mitigate reward hacking using a variational information bottleneck objective. Their experiments show the potential that their method might be an alternative to KL divergence for preventing reward hacking.

**Strengths:**

Reward hacking is an important problem in the field that is worth investigating.
Using an information bottleneck objective is an interesting idea. The derivation of the computable objective is insightful.

**Weaknesses:**

Although the method sounds interesting and neat, the experimental evaluation seems to have several issues that make it difficult to evaluate the practical benefit of the proposed method.

- The study is motivated by a phenomenon called reward overgeneralization. It does not provide any evidence of solving reward overgeneralization. Length bias is mentioned as an example, but it is not evaluated in the paper.
- Why the variational information bottleneck reduces the spurious correlation is not discussed in the paper. It’s not clear to me why it would be better than a normal LLM-based RM.
- The experimental results show that it outperforms RM without KL penalty in Figures 7-13. We already know that RM without KL penalty is prone to reward hacking and KL penalty is one of the solutions. Still, the evaluations in the Appendices compare InfoRM against this weakened baseline.

**Questions:**

- I would like to see the performance of KL regularized PPO with KL penalty larger than 0.01, say 0.1. Given that RM+KL is only marginally better than RM in Figure 4, I wonder if a larger KL penalty would improve the performance of RM+KL.
- It would be a great addition to the paper if RM+KL are compared in Figures 7-13.
- Does the proposed method solve the length bias problem? It was implied in the Introduction that it is one of the overgeneralization phenomena that the proposed method may solve.
- How is ensemble RM implemented?
- line 716: IB dimensionality is set to 3. Should we interpret it that ultimately the reward of the outputs can be explained by just three real numbers?
- Appendix D.2. Figure 15: Wouldn't removing ties make the estimation of the win rate worse than including ties as 0.5 wins? At least it would be informative to report the tie rate if it is removed from Figure 15.

**Limitations:**

I don't see any problems.

---

> ### Author Rebuttal · Authors · 2024-08-05
>
> We appreciate your positive feedback on the use of an information bottleneck objective and your recognition of the insightful derivation of the computable objective. We will address each of your comments and concerns below and also in our revised manuscript.
>
> ---
>
> > **W1**: Evidence of solving reward overgeneralization, such as length bias.
>
> **RW1:** Thanks for your comment. In fact, **we have discussed the role of our method in solving reward overgeneralization in Appendix C and cited it in Line 64, Introduction of our paper**. In this section, we demonstrate the significant impact of our method in mitigating length bias across twelve datasets. Furthermore, we also discuss other reward overgeneralization phenomena that our method can effectively mitigate, such as excessive caution.
>
> > **W2:** Why does our method reduce the spurious correlation?
>
> **RW2:** As stated in the Methodology section of our paper, by using variational information bottleneck, **our InfoRM is trained to maximize the utility of the latent representation for reward prediction while minimizing the irrelevance of human preferences within it**. This process eliminates the irrelevant information (i.e., spurious correlations), resulting in superior reward modeling as compared with the standard LLM-based RM, especially in alleviating reward overgeneralization, as verified in Appendix C of our paper. We will clarify this point further in the revised version.
>
> > **W3:** Comparison with a weakened baseline in Figures 7-13.
>
> **RW3:** Thanks for your feedback. Due to space constraints, we illustrate the **CSI values across various RMs (including RM+KL with differing KL penalties) during the RLHF processes using the AlpacaFarm, Anthropic-Harmless, and Anthropic-Helpful datasets in Figure 2 of the submitted PDF**. Additionally, the corresponding RLHF performance, as evaluated by GPT-4, is listed in Table 2 of the submitted PDF. Our results reveal the following:
>
> 1. As the KL penalty increases, the growth trend of CSI values is gradually suppressed, indicating a less pronounced reward over-optimization phenomenon. **This observation aligns with our intuition and further demonstrates the effectiveness of our CSI metric in detecting reward over-optimization**.
> 2. Although RM+KL achieves comparable performance to InfoRM in mitigating over-optimization, InfoRM consistently outperforms RM+KL in final RLHF performance. **This demonstrates that our method can significantly suppress the reward over-optimization phenomenon without largely compromising the final RLHF performance.**
>
> We will include relevant results on all testing datasets in the revised version.
>
> > **Q1**: Whether a KL penalty larger than 0.01 improve the performance of RM+KL in the simulated experiments?
>
> **RQ1**: To address your concern, we provide the simulated RLHF results for RM+KL with different KL penalty values, which can be found in Figure 1(a) of the submitted PDF. **Our findings indicate that a KL penalty value of 0.001 yields the best performance. When the KL penalty exceeds 0.01, the RLHF performance significantly degrades.** We will include relevant discussion in the revised version.
>
> > **Q2:** It would be a great addition to the paper if RM+KL are compared in Figures 7-13.
>
> **RQ2:** Please see the response to W3.
>
> > **Q3:** Does the proposed method solve the length bias problem?
>
> **RQ3:** Please see the response to W1.
>
> > **Q4:** How is ensemble RM implemented?
>
> **RQ4:** Ensemble RM in our experiments is implemented by combining the average reward across all models in the ensemble with the intra-ensemble variance, strictly following the UWO implementation in [1]. We will include this detail in the revised version.
>
> > **Q5:** Question about “IB dimensionality is set to 3." in Line 716.
>
> **RQ5:** We apologize for this typo. As analyzed in Appendix D of our paper, **the IB dimensionality in our experiments is set to 128, indicating that the final reward can be represented by a vector of this length**. We will correct this typo and double-check our paper.
>
> > **Q6:** Report the win rate considering ties in Figure 15.
>
> **RQ6:** Thanks for your feedback. In Figure 15 of our paper, our calculation of the win rate closely follows [2]. Following your suggestion, **we also report the win rate considering ties in Figures 1(b) and 1(c) in the submitted PDF**. We will include these results in the revised version.
>
> [1] Coste, Thomas, et al. "Reward Model Ensembles Help Mitigate Overoptimization." ICLR 2024.
>
> [2] Li, Yuhui, et al. "RAIN: Your Language Models Can Align Themselves without Finetuning." ICLR 2024.

---

> > ### Comment · Reviewer_rGMr · 2024-08-11
> >
> > Thank you very much for the clarification.
> > Now I think the contribution of the paper is clear and the empirical results it brings are interesting for a wide range of audiences.

---

> ### Author Response · Authors · 2024-08-14
> **Thanks for your response!**
>
> Dear Reviewer rGMr,
>
> Thank you very much for your positive feedback. We appreciate your recognition of the paper's contributions and the significance of our empirical results to a broad audience.
>
> Best regards.

---

### Author Rebuttal · Authors · 2024-08-06

Dear all Reviewers,

Thank you for your effort in reviewing our paper. The submitted PDF file includes the tables and figures referenced in our responses to your comments. The main contents of this PDF are listed as follows:

- Table 1 presents **the comparison results of RLHF models using different RMs, including a recently proposed method WARM and an extra dataset RL; DR Summary**, demonstrating the superiority of our InfoRM.

- Table 2 presents **the results of different hyperparameter settings for all compared methods**, ensuring the fairness and reliability of our experiments.

- Table 3 presents **the improvement in RLHF performance brought by using the proposed CSI metric as an early stopping method**, demonstrating the effectiveness of our CSI metric.

- Table 4 presents the accuracy of Standard RM and our InfoRM on in-distribution and out-of-distribution testing datasets, demonstrating that our InfoRM achieves better accuracy and generalization.

- Figure 1(a) presents **the simulated RLHF results for Standard RM with varying KL penalty values**, as well as for our proposed InfoRM, further demonstrating the superiority of our method.

- Figures 1(b) and 1(c) present **the parameter sensitivity analysis of our InfoRM, where the win rate is calculated considering ties**.

- Figure 2 presents **the proposed CSI metric values during the RLHF processes of Standard RM with varying KL penalty values**, as well as our InfoRM, validating the effectiveness of our CSI metric for detecting reward overoptimization.

Thanks for your time!

Sincerely, Paper 9339 Authors.

---

### Decision · Program_Chairs · 2024-09-25

**Decision:**

Accept (poster)

**Comment:**

This paper proposes an information theoretic objective for reward modelling for RLHF. In a nutshell, the method maximises the mutual information between the latent space and the reward model prediction, while minimising the spurious (useless) MI between the input and the latent space when conditioning on a given output.

The results overall seem convincing, both in terms of the evaluation on a gold reward model and in head-to-head based on GPT4 evaluation. While I am generally quite sceptical of "gold reward models" (I believe they also get hacked), I think the head-to-head results are convincing.

While this is a borderline paper I am currently leaning "accept" on balance.

It would be great if the authors could address why the _trivial_ solution (latent == output) doesn't just happen.

I also ask the authors to make sure they stick to the page limits for the camera-ready-copy.